# Position: Significant impact of numerical precision in scientific machine learning

## Abstract

The machine learning community has focused on computational efficiency, often leveraging reduced-precision formats such as those below the standard FP32. In contrast, little attention has been given to higher-precision formats, such as FP64, despite their critical role in scientific domains like materials science, where even small numerical differences can lead to significant inaccuracies in physicochemical properties. This need for high precision extends to the emerging field of *machine learning for scientific tasks*, yet it has not been thoroughly investigated. According to several studies and our toy experiment, models trained with FP32 exhibit insufficient accuracy compared to FP64, indicating that higher precision is also crucial in scientific machine learning, as it is in traditional scientific computing. Despite the potential of scientific machine learning, this precision issue often limits its adoption as replacements for traditional scientific computing in practical research. This position paper not only highlights these precision-related issues but also recommends reporting comparisons between FP32 and FP64 results while encouraging the release of FP64 models. We believe these efforts can enable machine learning to contribute meaningfully to the natural sciences, ensuring both scientific reliability and practical applicability.

## 1. Introduction

The rapid advancements in natural language processing (NLP) and computer vision (CV) in the machine learning (ML) field have accelerated the broad application across various domains (Litjens et al., 2017; Ozbayoglu et al., 2020; Ren et al., 2021; Lai et al., 2024; Raghu & Schmidt, 2020). Specifically, **ML for scientific tasks**–*which has begun to re-solve intellectually demanding problems in scientific fields*– has been highlighted across disciplines, opening new possibilities for scientific breakthroughs. In recognition of these breakthroughs, the 2024 Nobel Prize in Chemistry honored the contributions of scientific ML, highlighting innovations such as AlphaFold and RoseTTAFold (Jumper et al., 2021; Baek et al., 2021; The Royal Swedish Academy of Sciences, 2024). These models transformed research by rapidly delivering results that once required significant resources and time-consuming experiments or simulations. Building on these successes, scientific ML not only addresses traditional labor-intensive workflows but also finds hidden patterns within complex data, thereby providing human researchers with direct insights into novel discoveries across natural sciences (Webb et al., 2018; Morgan & Jacobs, 2020; Karagiorgi et al., 2022; Yoo et al., 2024).

In the context of methodology, the development of scientific ML naturally follows the broader trends and paradigms of the ML research field. In the early stages of NLP and CV, most work focused on discriminative tasks (*e.g.*, named entity recognition and image classification) (Walker et al., 2006; Deng et al., 2009) before gradually shifting to generative tasks (*e.g.*, machine translation and text-to-image generation) (Bojar et al., 2014; Schuhmann et al., 2022). Further, generative approaches themselves have advanced sequentially, moving from variational autoencoders (VAEs) to generative adversarial networks (GANs), and more recently, to diffusion models (Kingma & Welling, 2014; Goodfellow et al., 2014; Ho et al., 2020; Song & Ermon, 2019). In a similar manner, numerous scientific domains have rapidly adopted the latest advances from the ML community. For example, among various areas of bioinformatics, research on DNA sequence data initially leveraged discriminative models such as DeepVariant (Poplin et al., 2018) and DeepSEA (Zhou & Troyanskaya, 2015), and over time, this expanded to generative models including ExpressionGAN (Zrimec et al., 2022) and Evo (Nguyen et al., 2024). Similarly, material structure prediction in the field of materials and drug discovery has followed this trend from VAEs (Sanchez-Lengeling & Aspuru-Guzik, 2018; Gómez-Bombarelli et al., 2018; Lim et al., 2018) and GANs (Prykhodko et al., 2019; Kim et al., 2020; Abbasi et al., 2022) to diffusion models (Hoogeboom et al., 2022; Peng et al., 2023; Zeni et al., 2025).

---

[1]Anonymous Institution, Anonymous City, Anonymous Region, Anonymous Country. Correspondence to: Anonymous Author <anon.email@domain.com>.

Preliminary work. Under review by the International Conference on Machine Learning (ICML). Do not distribute.

In parallel with these advances, the most recent paradigm in ML research, often referred to as the *scaling law*, focuses on improving performance by progressively increasing the size of models, training datasets, and computational resources (Kaplan et al., 2020; Snell et al., 2024). Building upon this idea of continuously expanding scale, researchers have successfully validated the approach across diverse fields, including NLP, CV, reinforcement learning, and time-series forecasting (Zhai et al., 2022; Cherti et al., 2023; Hilton et al., 2023; Neumann & Gros, 2023; Shi et al., 2024). Accordingly, the scientific ML domain will also adopt this paradigm, and in fact, large models designed to address scientific tasks have already begun to emerge (Nguyen et al., 2024; Zhang et al., 2024).

As these models grow larger and more complex, they inevitably require massive computational power, which results in a significant challenge for both training and inference. To address this, the lower numerical precision, or quantization is a widely employed strategy, which helps reduce the computational expense (Zhu et al., 2024; Micikevicius et al., 2018). These approaches inevitably involve a trade-off between accuracy and computational budgets, resulting in an unwanted loss of accuracy. To minimize such losses, techniques such as mixed precision training (Micikevicius et al., 2018) or more advanced quantization methods (Banner et al., 2019; Dettmers et al., 2022; Liu et al., 2023; Xu et al., 2024) have been proposed, which allow researchers to conserve the original accuracy while achieving the advantages of reduced computational costs. Consequently, the ML community has accepted slight accuracy degradation as a natural trade-off for greater efficiency, thereby integrating these lower-precision techniques into real-world applications to balance computing cost and performance.

However, the tolerance for lower-precision techniques is highly problematic in the field of scientific computing. Scientific computing primarily aims to solve fundamental physics equations that are difficult to solve manually by simplifying or discretizing the inherently continuous and infinite real-world phenomena to make them computationally tractable. As a consequence, even tiny differences in numerical precision can lead to significant issues regarding the reliability of computational results. Specifically, our experimental results show that *single precision's sensitivity to numerical deviations can substantially influence the accuracy of fundamental physical equations*. As a result of this high sensitivity, small precision differences can cause significant changes in physicochemical properties, such as absorption coefficient, defect energies, or reaction pathway predictions, thereby reducing the reliability of results, especially when accurate predictions are crucial for critical decisions. One critical aspect is that these challenges related to numerical precision are not confined to traditional computational science, as ML models are increasingly being utilized in various studies to replace prevalent simulations. In other words, traditional computational science requires high precision, making it essential to verify whether FP32 produces valid results before using ML models, as numerical precision is key to maintaining reliability.

**In this position paper, we argue for the significant role of numerical precision in scientific ML research, emphasizing the need for evaluating and analyzing its impact on results derived from varying precision levels.** To this end, we first present real-world examples from previous computational simulations where numerical precision had a notable impact on their results. By introducing cases that reflect real-world scenarios from actual research fields, we aim to demonstrate the practical existence of numerical precision issues in scientific simulations. Subsequently, we explain that the importance of numerical precision is not confined to traditional scientific computing alone but is also deeply related to ML applications in scientific domains. Specifically, we provide examples involving ML potential models and physics-informed neural networks (PINNs), which are actively studied in both ML and science domains, demonstrating the critical role of numerical precision in these areas (Raissi et al., 2019; Kocer et al., 2022; Käser et al., 2023). Furthermore, we address the growing use of large language models (LLMs) in scientific ML and their implications for precision-related challenges.

In conclusion, we propose actionable recommendations for the ML community and potential research directions based on our earlier discussions. We then present alternative viewpoints, offer responses, and conclude. Since the main role of ML in scientific research is to deepen understanding in traditional domains, the issues we raise must be rigorously examined. When relatively simple actions by ML researchers can remove barriers that hinder natural scientists from applying ML models, these measures become essential, not optional. As scientific machine learning is still in its early stages, we hope that thorough debate will help minimize trial-and-error in future research.

## 2. Importance of numerical precision in scientific computing

In scientific computing, the main goal is solving complex physics equations through computational power, especially when manual solutions are impractical or nonexistent. Many-body problems including multiple object interactions demonstrate the necessity of high-performance computing solutions. Various computational methods have emerged to solve fundamental physics equations: molecular dynamics for Newton's Second Law, density functional theory (DFT) (Jones & Gunnarsson, 1989) for the Schrödinger equation, and the finite-difference time-domain (FDTD) (Yee, 1966) method for Maxwell's equations.

Modern digital computers use discrete bit-based representations, creating an inherent challenge. The floating-point system limits direct solutions to physical equations that operate in continuous systems, including $F = ma$, Schrödinger equation, and Maxwell's equations. Researchers have developed workarounds using approximated equations (*e.g.* the Kohn-Sham equation (Kohn & Sham, 1965)) or discretization methods with specific time units and grid systems (*e.g.* molecular dynamics). These solutions require high numerical precision to accurately represent physical phenomena, typically settling on double precision as a balance between computational cost and accuracy. For instance, Quantum ESPRESSO (Giannozzi et al., 2009), a leading open-source DFT implementation, strictly enforces double precision throughout its code.

To demonstrate the precision's crucial role in scientific computing, we present examples showing how small numerical variations can significantly impact computational results, analyzing these effects in realistic research scenarios. Specifically, we illustrate the influence on realistic research scenarios, thereby analyzing the implications and identifying the precise numerical accuracy-related challenges.

## 2.1. Impact on density functional theory simulation

Quantum mechanics, beginning with Planck's quantum hypothesis (Planck, 1900), revolutionized our understanding of microscopic phenomena. While exact calculations are only possible for simple systems like the hydrogen atom, the Kohn-Sham equation introduced density functional theory (DFT) as an efficient approach for many-body electron problems. Using Python-based Simulations of Chemistry Framework (PySCF), we performed geometry optimization calculations for water ($H_2O$) using both Hartree-Fock (HF) and DFT calculations with B3LYP functional and 6-311++G(d,p) basis set (Andersson & Uvdal, 2005; Tirado-Rives & Jorgensen, 2008; Yanai et al., 2004).

Figure 1 shows the results of geometry-optimized water molecules obtained from HF and DFT calculations under different numerical precision conditions: FP32 and FP64. When utilizing FP64, both HF and DFT calculations successfully converged within three optimization steps with satisfying the convergence criteria. Since DFT explicitly accounts for electron correlation effects (Becke, 1988), it is generally expected to provide more accurate results than HF, a trend that is also reflected in our findings. Comparing bond lengths, the reference (Bowen & Sutton, 1958) O-H bond length is 0.957 Å, while HF exhibits a deviation of 0.016 Å (1.7 % error), and DFT yields a smaller deviation of 0.005 Å (0.5 % error). Similarly, for the bond angle, HF deviates by 1.7 °(0.7 % error) from the reference value of 104.52 °, whereas DFT shows a smaller deviation of 0.55 °(0.5 % error). However, when using FP32, significant numerical

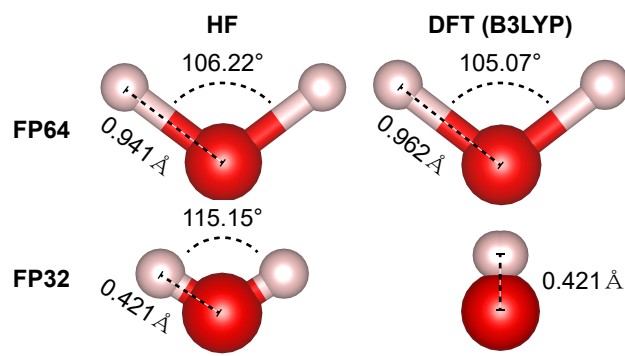

*Figure 1.* Water molecule geometry optimizations comparing FP64 (top) and FP32 (bottom) using HF (left) and DFT (right) methods. FP64 yields physically valid structures while FP32 produces unrealistic geometries.

*Table 1.* Atomic coordinates and total energy comparison of geometry-optimized $H_2O$ at FP32 and FP64 precision using 6-311++G(d,p) basis set. (*) FP32 failed to converge in both HF and DFT methods, while FP64 showed consistent results.

|  |  | **HF** | | **DFT** (B3LYP) | |
|---|---|---|---|---|---|
|  |  | 6-311++G(d,p) | | 6-311++G(d,p) | |
|  |  | FP32 | FP64 | FP32 | FP64 |
| Atomic coordinates (Å) | $O_x$ | -0.000356* | 0.000000 | 0.009524* | 0.000000 |
|  | $O_y$ | 0.246311* | 0.014028 | 0.578655* | 0.000780 |
|  | $O_z$ | 0.000000* | 0.000000 | 0.000000* | 0.000000 |
|  | $H_1x$ | 0.453099* | 0.752792 | 0.026584* | 0.763642 |
|  | $H_1y$ | 0.534244* | 0.578999 | 0.998814* | 0.585902 |
|  | $H_1z$ | 0.000000* | 0.000000 | 0.000000* | 0.000000 |
|  | $H_2x$ | -0.453725* | -0.752792 | -0.024889* | -0.763642 |
|  | $H_2y$ | 0.534404* | 0.578999 | 0.998054* | 0.585902 |
|  | $H_2z$ | 0.000000* | 0.000000 | 0.000000* | 0.000000 |
| Total energy (Ha) |  | -74.938* | -76.053 | *N/A* * | -76.458 |

*\*Not Converged*

instabilities arise, preventing the convergence of optimization steps. In the case of HF calculations, the gradient of hydrogen atoms stagnates between 0.2–0.4 Ha/Bohr, which is significantly above the desired convergence threshold of $10^{-6}$ Ha/Bohr. For DFT calculations, the issue becomes even more pronounced, as the gradient values rapidly diverge beyond $10^5$ Ha/Bohr, resulting in termination before reaching the maximum step. As a result, when using FP32, the HF calculation exhibits a substantial 50 % error, while the DFT calculation produces a molecular structure impossible to exist in reality, as illustrated in Figure 1.

A detailed examination of the atomic coordinates in Table 1 further highlights the differences. While the coordinates obtained from FP64 differ only by approximately 0.01 Å for oxygen and hydrogen atoms, FP32 results display considerable deviation. Notably, the FP32-calculated atomic positions deviate by up to 0.4 Å from those obtained using FP64, a significant difference considering that the O-H bond

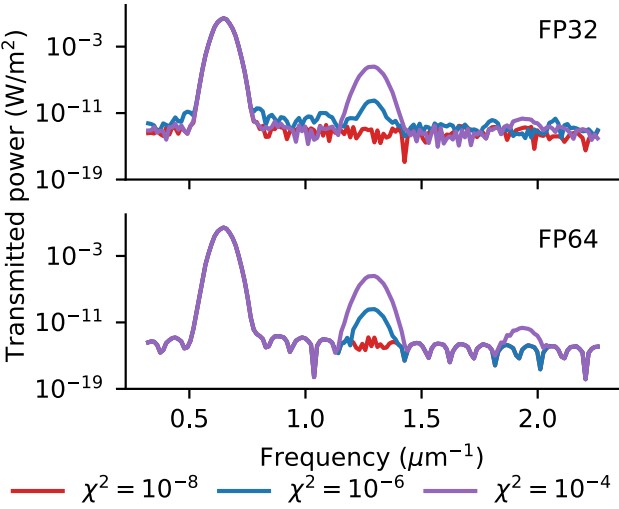

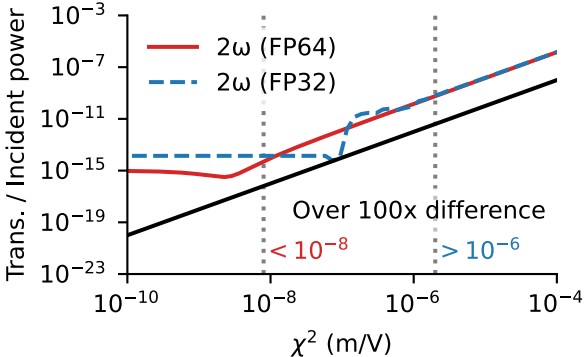

*Figure 3.* Computed second harmonic susceptibility shown in FP64 (red) and FP32 lines (blue dashed) compared to theoretical quadratic behavior (black). FP64 maintains accuracy to $10^{-8}$ m/V, while FP32 deviates above $10^{-6}$ m/V, rendering it unsuitable for typical nonlinear materials.

*Figure 2.* Transmittance spectra comparison between FP32 (top) and FP64 (bottom) in Kerr media, showing FP32's failure to accurately model higher harmonics and low-power wave patterns below $10^{-10}$ W/m$^2$.

length itself is only 0.957 Å. In addition, the total energy difference between FP32 and FP64 calculations is approximately 1.1 Hartree (equivalent to 29.93 eV), which exceeds the formation energy of water (2.9 eV) by more than an order of magnitude. This clearly indicates that the FP32 result corresponds to a structure that is impossible to exist in reality. These results demonstrate that FP32 lacks the numerical precision necessary to achieve sufficient convergence tolerance in scientific computations. The failure of a simple molecular system such as water to reach an optimized structure under FP32 precision indicates its fundamental limitations in scientific calculations.

### 2.2. Impact on finite difference time domain simulation

Electromagnetism, established by Maxwell's equations (Maxwell, 2010; 1865), provides the theoretical foundation for understanding electromagnetic waves. However, solving Maxwell's equations for complex phenomena is computationally challenging. To address this, FDTD discretizes Maxwell's equations in time and space. Using Meep (Oskooi et al., 2010), which an open-source FDTD software, we investigated numerical precision effects on electromagnetic simulations, comparing FP32 and FP64 in nonlinear Kerr media simulations. We simulated a Kerr medium (refractive index=1.65) excited by an electromagnetic wave source ($\lambda$=1.55 $\mu$m, $\Delta\lambda$=0.15 $\mu$m).

Figure 2 presents the transmission spectrum of the nonlinear Kerr medium under FP32 and FP64 precision settings. From left to right, the spectral peaks correspond to the fundamental generation induced by the source, the second

harmonic generation (SHG), and the third harmonic generation (THG). While the fundamental peak exhibits minimal differences between FP32 and FP64, notable discrepancies arise in the SHG and THG regions. Specifically, FP32 calculations display pronounced background signal instability and intensity variations in harmonic generation, which result from imprecise numerical computation. A particularly notable difference appears in the behavior of the background signal. In FP64 calculations, the background follows a well-defined periodic pattern governed by the electromagnetic wave, whereas in FP32, the background signal appears as unstructured Gaussian-like noise. This phenomenon indicates that the lack of numerical precision in FP32 significantly disrupts the accurate computation of low-intensity transmitted power, particularly for electromagnetic waves in the range of $10^{-11}$ W/m$^2$. These findings highlight the fundamental limitations of single precision (FP32) in reliably capturing weak electromagnetic signals and nonlinear optical effects.

To further analyze the impact of numerical precision, we examined the relationship between second-order nonlinear susceptibility ($\chi^2$) and the transmittance-to-incident power ratio. As shown in Figure 3, the black upward-sloping line represents a quadratic line, serving as a reference line indicating the expected computational trend of transmittance over incident power ratio as nonlinear susceptibility varies. Ideally, the computationally simulated values should align with this reference trend, maintaining the same slope. Comparing the results obtained from FP64 (red solid line) and FP32 (blue dashed line), we observe that as nonlinear susceptibility decreases beyond a certain threshold, the ratio begins to saturate. This saturation point effectively defines the lower bound of computational precision achievable under each numerical setting.

Specifically, for values of $\chi^2$ above $10^{`6}$, both FP64 and FP32 provide reliable computational precision. However, for values below this threshold, FP32 results begin to exhibit saturation, rendering further calculations meaningless due to the loss of numerical resolution. In contrast, FP64 maintains simulation accuracy down to approximately $10^{-8}$, demonstrating a computational precision that is at least two orders of magnitude higher than that of FP32. This result implies that for most nonlinear materials with $\chi^2$ values below $10^{-6}$, transmittance spectrum simulations using FP32 become inherently unreliable. These findings highlight the critical role of numerical precision in computational science, particularly in fields where small numerical deviations can lead to substantial errors. As demonstrated in both DFT and FDTD simulations, the limitations of single precision introduce significant inaccuracies, especially in cases involving highly sensitive physical properties. This also highlights the necessity of carefully selecting numerical precision levels when conducting computational simulations–particularly in scientific ML applications–where maintaining the reliability of results is essential.

## 3. Numerical precision issue in scientific ML

As demonstrated in the previous section, numerical precision can significantly affect the outcomes of traditional scientific simulations and potentially influence the results of scientific research. This naturally leads to an important question: **Do ML models designed for scientific tasks also suffer from similar precision-related issues?** To investigate this question, we survey various studies that apply ML to scientific research, searching for cases where the precision issue has been reported. We also conduct simple toy experiments to further assess the impact of numerical precision in ML-based scientific tasks. Through these analyses, we seek to determine whether the precision issue is a *significant challenge* or *just a theoretical concern*.

### 3.1. Impact on machine learning potential

The first example we present is an ML potential[1] (Kocer et al., 2022; Käser et al., 2023), which is closely related to Section 2.1. Fundamentally, ML potential models aim to compute potential energy and the associated forces for a given material structure, offering a much faster alternative to traditional quantum mechanical calculations. Due to their wide range of applications, ML potentials have been extensively studied not only in physics, materials science, chemistry, and biology but also within the ML community (Behler & Parrinello, 2007; Pukrittayakamee et al., 2009; Smith et al., 2017; Gilmer et al., 2017; Schütt et al., 2017). In addition, closely related topics, such as property

---

[1]In other domains, the term *machine learning interatomic potential* (MLIP) is also used.

prediction and generation for material or drug discovery have also been actively explored, making ML potentials a familiar subject for ML researchers. In this position paper, we specifically focus on ML potentials based on neural networks, *i.e.*, neural network potentials. Since ML research often treats energy and force values in the same manner as other material properties, our discussion extends naturally to broader property prediction tasks.

A key challenge in ML potential studies lies in effectively representing and processing atomic information in three-dimensional space while ensuring rotational and translational equivariance or invariance. To tackle this, the field has evolved from vanilla graph neural networks (Scarselli et al., 2009) and transformers (Vaswani et al., 2017) to more specialized architectures that satisfy these constraints, achieving higher prediction accuracy (Schütt et al., 2017; Gasteiger et al., 2021; Satorras et al., 2021; Batzner et al., 2022; Batatia et al., 2022; Fuchs et al., 2020; Thölke & Fabritiis, 2022; Liao & Smidt, 2023). As a result, many recent models are now integrated into widely used libraries or simulation software, such as the Atomic Simulation Environment (ASE) (Bahn & Jacobsen, 2002; Larsen et al., 2017) and LAMMPS (Plimpton, 1995; Thompson et al., 2022). This demonstrates that ML potential models are increasingly employed in practical research; thus, any numerical precision issues arising in these models could have significant implications for scientific discoveries.

Consequently, we aimed to investigate whether existing ML potential models suffer from precision issues. To this end, we surveyed the pretrained checkpoints of various ML potential models available in the ASE library to determine whether they support FP64 precision. Interestingly, among several models in ASE, only MACE (Batatia et al., 2022) provides pretrained checkpoints trained in FP64, while other models appear not to have considered FP64 training. Even before detailed analysis, this observation suggests that the ML potential community may not be fully aware of the potential significance of numerical precision.

To preliminarily understand the effect of precision, we conducted a toy experiment using MACE, the only model that provides FP64-trained parameters. We selected an ethanol molecule as a manageable small organic system containing multiple atom types (C, H, O), then moved one of its carbon atoms (specifically the one closest to oxygen) along a certain path (shown in the top of Figure 4) and observed changes in the potential energy and forces. We compared the results obtained in FP32 with those in FP64 by applying built-in type conversion in the MACE code to the FP64-trained checkpoint. The upper plot in Figure 4 illustrates the overall trends in energy and force, while the lower plot shows the direct differences between FP32 and FP64. Our results show that the differences remain within approximately 1

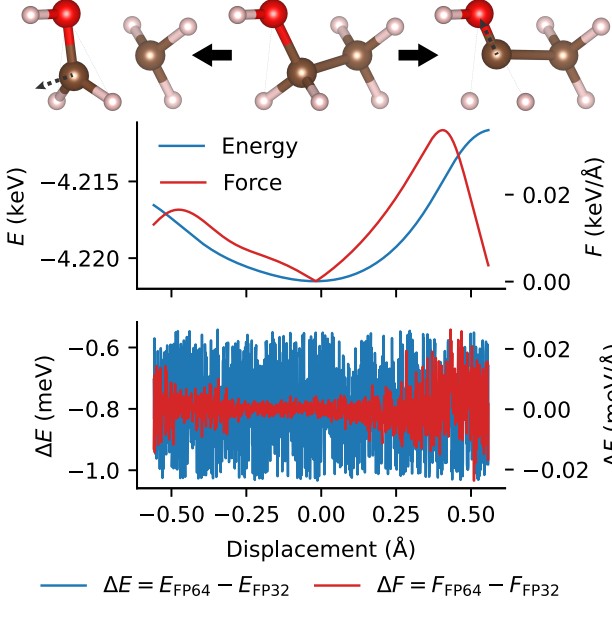

*Figure 4.* MACE model calculations showing energy (red) and force (blue) changes during carbon atom displacement in ethanol. FP32 versus FP64 precision reveals minimal deviations (1 meV energy, 0.02 meV/Å force), though broader testing needed.

meV for energy and 0.02 meV/Å for force, which is often considered acceptable in small-molecule simulations.

However, this experiment was deliberately simplistic, and the results should not be overinterpreted as evidence that FP32 models are generally reliable. Larger or more complex systems, such as polymers or proteins, could exhibit more substantial numerical errors. Additionally, the FP32 model tested here was originally trained in FP64 and then converted to FP32 for inference; a model trained entirely in FP32 from the outset might behave differently. In fact, Batatia et al. (2025) report that NequIP (Batzner et al., 2022) exhibits different numerical sensitivity when trained in FP32 versus FP64, and Maxson et al. (2024) also discuss similar issues. These observations highlight the importance of carefully assessing numerical precision in ML potential models and the need for systematic benchmarks regarding precision.

### 3.2. Impact on physics-informed neural network

Beyond the fundamental equations mentioned in the previous section, various subfields of natural science describe natural phenomena using differential equations. For example, in fluid dynamics, including weather prediction, Navier-Stokes, continuity, and heat transfer equations are commonly used (Tritton, 2012; Bauer et al., 2015). Moreover, differential equations such as the Black-Scholes equation (Black & Scholes, 1973) are also employed in fields beyond natural sciences, such as financial engineering. Many of these equations either lack general analytical solutions

or are too complex to be solved manually. As a result, numerical methods have been developed over time, leading to techniques such as the Euler method, Runge-Kutta methods, and Picard method (Butcher, 2016; Strauss, 2007). These techniques have also influenced modern approaches in ML, including diffusion models, NeuralODEs, and deep equilibrium models (Ho et al., 2020; Song & Ermon, 2019; Chen et al., 2018; Bai et al., 2019).

The concept of the PINNs (Raissi et al., 2019) leverages automatic differentiation (autograd), fundamental to backpropagation, to solve differential equations using neural networks. Due to its simple yet powerful approach, PINNs have been widely adopted in scientific domains that rely on numerical methods. This section explores whether numerical precision issues also arise in PINNs and investigates related challenges through a literature survey.

First, Nakamura et al. (2022) explicitly discussed the impact of numerical precision in scientific research, reporting that training PINNs with FP32 failed, whereas FP64 did not: *from a comprehensive standpoint, FP32 computation has a risk of failure for the present problem compared with FP64.* This work applies PINNs to a specific fluid dynamics problem involving surface tension modeling, which requires up to fourth-order derivatives, making it a specialized case of differential equations. Although this is a specific scenario, it is a real-world scientific study, demonstrating that precision issues can significantly impact the practical use of PINNs.

Meanwhile, Sharma & Shankar (2022) were well aware of precision issues and leveraged this understanding to improve the methodology of PINNs. The key idea of their work is to replace certain autograd operations in PINNs with a specialized FD method, reducing the computational cost associated with autograd. Here, to compensate for the loss of accuracy introduced by finite difference approximations, the authors proposed using high-precision (FP64) training. As a result, the reduction in computational cost from bypassing autograd exceeds the overhead introduced by FP64 operations, leading to an overall speedup that makes their approach faster than a vanilla PINN in FP32. Beyond the fields of PINNs and scientific ML, this study introduces a novel perspective on utilizing high-precision models in neural network research.

Thus, in the context of PINNs, a comprehensive study is needed to systematically assess the impact of numerical precision issues on scientific research. Fortunately, many fields share similar types of differential equations, *e.g.*, Laplace equation in electrostatics and fluid dynamics, where it describes electric potential distribution and velocity potential in inviscid flow, respectively. By focusing on the precision challenges of commonly used differential equations and rigorously validating PINNs in this context, such research could have a substantial impact across multiple domains.

### 3.3. Challenges for large language models

The emergence of LLMs in scientific applications is accelerating, further raising concerns about numerical precision in such domains. To investigate these concerns, we examine both existing studies and empirical evidence that highlight precision-related challenges in LLM applications. The integration of LLMs in scientific domains follows two distinct approaches. The first involves direct inference without architectural modifications, where scientific data is transformed into natural language format for existing LLM architectures (Rubungo et al., 2024; Jacobs et al., 2024; Liu et al., 2024). The second approach develops specialized architectures that combine domain-specific encoders with fine-tuned language models, preserving the intrinsic properties of scientific data while leveraging LLM capabilities (Li et al., 2024; Park et al., 2024).

Regarding the first approach, unlike conventional scientific models, LLMs generate outputs based on tokens, which may compromise prediction accuracy. Numerous studies have demonstrated that LLMs struggle with symbolic tasks (Wei et al., 2022; Yao et al., 2023), similar to their difficulties in numerical predictions. For instance, these models often fail to accurately count the occurrences of specific characters within words (*e.g.*, counting the letter 'r' in 'strawberry') or comparing the size of decimal numbers (*e.g.*, determining whether 3.9 is larger than 3.11[2]). This limitation stems from their fundamental architecture, where words are processed as sequences of tokens rather than as individual alphabetic characters or numbers. Although various studies (Wei et al., 2022; Kojima et al., 2022; Yao et al., 2023; Besta et al., 2024) have been proposed to address these challenges, symbolic manipulation remains a significant obstacle for LLMs. Consequently, their application in scientific tasks requires careful consideration and validation.

Another critical consideration in LLM deployment is the continuous increase in model size. For instance, the open-source Llama series demonstrates this trend clearly: LLaMA (65B parameters) grew to Llama-2 (70B) and further to Llama-3.1 (405B) (Touvron et al., 2023a;b; Llama Team, 2024), and more recently, DeepSeek-v3 has pushed this expansion even further, reaching 671B (DeepSeek-AI, 2024). Such explosive growth in model sizes across LLMs has resulted in a substantial increase in computational costs for both training and inference. To mitigate the budget, researchers commonly employ parameter quantization techniques by reducing model precision to lower-bit formats (Liu et al., 2021; Dettmers et al., 2022; Liu et al., 2023), sometimes even 1-bit representations (Xu et al., 2024).

---

[2]In January 2025, GPT-4o incorrectly answers that *3.11 is larger than 3.9*, due to tokenization: 3.9 as ['3', '.', '9'] and 3.11 as ['3', '.', '11'], leading to a direct comparison of 9 and 11.

However, these approaches directly contradict the high precision requirements of scientific computing, which we have emphasized throughout this discussion. This issue is particularly critical for the second approach, where domain-specific encoders—often derived from scientific ML models—serve as feature extractors. If quantization significantly reduces the precision of the extracted features, the LLM may fail to process them accurately, potentially degrading overall model performance. For example, Li et al. (2024) employed UniMol (Zhou et al., 2023), a model broadly categorized as an ML potential, as an encoder. Even if the encoder provides highly precise features, the LLM's lower precision representations may obscure this information, ultimately leading to inaccurate final predictions. This inherent trade-off between computational efficiency and numerical precision highlights the importance of carefully designing LLM integration strategies in scientific applications to ensure both accuracy and practicality.

## 4. Suggestions for Advancing Scientific ML

Building on previous discussions, we outline key directions for the ML community to advance scientific tasks.

**Exploring high-precision models and mixed high-precision training**  Most ML research has primarily explored lower-precision formats such as FP16, BF16, and INT8, whereas comparisons between FP32 and FP64 remain relatively limited. We argue that *researchers should explore the potential benefits of double precision beyond the commonly used FP16 and FP32*. Inspired by mixed-precision training, we propose extending this idea to high-precision training. Similar to conventional mixed-precision training, which employs lower precision for the majority of layers while preserving higher precision for numerically sensitive operations (*e.g.*, batch normalization and softmax), we propose identifying sensitive layers and selectively training them in FP64. This direction is especially relevant from an energy efficiency perspective, as FP64 training inherently consumes more energy than FP32. While scientific ML is often considered advantageous over traditional scientific computing in terms of runtime, its energy consumption remains a critical concern. Investigating novel model architectures and training techniques that preserve high numerical precision while enhancing energy efficiency will be crucial for the widespread adoption of scientific ML.

**Benchmarking and reporting FP32 vs. FP64 results**  Scientific ML typically demands higher numerical precision than general ML applications to ensure computational reliability. While predictive accuracy is the primary focus, other factors such as training and inference time remain significant , as well as energy efficiency. Consequently, researchers should explicitly report the numerical preci-

sion used in their studies, compare FP32- and FP64-trained models where applicable, and publicly release FP64-trained models to enhance reproducibility and facilitate further advancements. To support meaningful evaluations, standardized benchmarks that capture precision sensitivity across diverse scientific tasks are essential. Such benchmarks would provide a consistent framework for quantifying trade-offs between numerical precision, computational efficiency, and reproducibility in scientific ML research.

**Collaboration with natural scientists** Achieving meaningful progress in scientific ML requires close collaboration with natural scientists. This is not merely a conceptual argument but a practical requirement, as ML researchers often lack the domain-specific intuition to determine the appropriate level of numerical precision for a given scientific task. For instance, research on ML potential is published in both traditional scientific journals and ML conferences, yet the evaluation criteria and priorities differ significantly between these communities (Batatia et al., 2022; Kovács et al., 2023). Strengthening interdisciplinary collaboration will help bridge this gap, ensuring that precision requirements align with both scientific validity and practical usability.

**Integrating ML into traditional computational methods** Rather than solely focusing on developing high-precision ML models, an alternative approach is to integrate ML into traditional computational methods to achieve both accuracy and efficiency. One promising strategy is to first employ ML models while acknowledging their inherent numerical limitations and using them to generate an approximate solution (Arisaka & Li, 2023; Saverio, 2023; Napier, 2024). These ML-generated approximations can then serve as an initial guess for traditional computational methods, significantly accelerating convergence while preserving precision. This hybrid approach presents a compelling solution for scientific applications where both computational speed and numerical accuracy are critical.

## 5. Alternative Views

This section presents alternative views that challenge our position and provides responses to address these concerns.

**Q1: Is the issue really about numerical precision, or could it be a capacity limitation of the model?** An alternative view holds that the observed inaccuracies stem from fundamental limitations in network architecture or training methods, rather than numerical precision constraints. According to this viewpoint, neural networks may not yet be sufficiently expressive to solve the given task, regardless of numerical precision constraints. To distinguish numerical issues from capacity concerns, we can leverage numerical analysis tools, such as condition numbers and numerical

sensitivity analysis, to determine whether errors arise from numerical instability. Since modern neural networks rely heavily on matrix operations, existing research on matrix sensitivity provides a robust foundation for further analysis. These insights can help clarify the relationship between numerical stability and model expressivity.

**Q2: If certain scientific computing tasks are not sensitive to numerical precision, does it matter?** It is true that not all scientific tasks require high numerical precision because certain tasks can tolerate lower levels of precision. However, our focus should be on fields where high precision is essential, such as quantum chemistry, materials science, and nonlinear physics, where even slight inaccuracies can lead to significant deviations. Currently, there is still limited understanding of which tasks, models, and environments are most affected by numerical precision and what factors contribute to these sensitivities. A systematic analysis is necessary to identify precision-critical cases before making broad assumptions about acceptable precision levels. Until a clear understanding is established, a precision-aware approach should be prioritized, while relaxed conditions can be applied to tasks that do not require high precision.

Moreover, certain scientific tasks may not require explicit consideration of numerical precision. For example, in tasks where logical reasoning is more critical than numerical accuracy, such as those that rely on LLMs, precision constraints may be less significant. These include (1) Explaining or summarizing experimental results or literature (Xie et al., 2024), (2) generating new hypotheses for scientific research (Kumbhar et al., 2025; Lu et al., 2024), (3) providing guidance for tasks where the methodology is not clearly defined (*e.g.*, retrosynethsis), and (4) assisting scientific educations (Bewersdorff et al., 2025). In such cases, the role of ML extends beyond numerical fidelity, focusing instead on knowledge synthesis and interpretability.

## 6. Conclusions

Scientific ML has become a major field in modern ML research, with the goal of developing models that contribute to scientific discovery. This position paper highlights the impact of precision issues, which can affect the practical usability of scientific ML models but have been largely overlooked. The precision issues in scientific ML are closely tied to ethical concerns regarding the reliability and explainability of scientific findings. In summary, our contribution lies in a practical step toward making scientific ML models more reliable, reducing the risk of misleading scientific insights due to numerical inaccuracies. If our simple yet easily actionable proposal becomes widely adopted in scientific ML research, it can enhance the practicality of models and thereby accelerate scientific discovery.

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

## A. Details in quantum mechanical calculation

Quantum mechanical calculations (also known as quantum chemistry calculations) were performed using PySCF (Python-based Simulations of Chemistry Framework, version 2.7.0). The input water molecule consists of a single oxygen atom at (0.000000, 0.000000, 0.000000) and two hydrogen atoms at (0.757000, 0.586000, 0.000000) and (-0.757000, 0.586000, 0.000000), respectively. All simulations were conducted using two nodes of an AMD EPYC 7543 32-Core Processor. To compare the geometry optimization result of a water molecule based on different exchange functionals, we performed both Hartree-Fock calculations and density functional theory calculations using the B3LYP functional. For both methods, we employed the 6-311++G(d,p) basis set. To evaluate the impact of numerical precision, we conducted the same calculations using both single precision (FP32) and double precision (FP64) by declaring np.float32 and np.float64, respectively. The default convergence tolerances for structural stabilization were set as follows: $|\Delta E| < 1.00 \times 10^{-6}$, RMS-Grad $< 3.00 \times 10^{-4}$, Max-Grad $< 4.50 \times 10^{-4}$, RMS-Disp $< 1.20 \times 10^{-3}$, and Max-Disp $< 1.80 \times 10^{-3}$.

## B. Finite-difference time-domain calculation

### B.1. Nonlinear Material Properties

In this study, Kerr media were modeled with a second-order nonlinear susceptibility ($\chi^2$) ranging from $10^{-12}$ to $10^{-2}$ and the refractive index was set to 1.65 to mimic the conventional nonlinear materials like beta barium borate. The nonlinear polarization of the material was expressed as:

$$P = \epsilon_0(\chi^{(1)}E + \chi^{(2)}E^2 + \chi^{(3)}E^3 + ...) \tag{1}$$

And the second-order nonlinear polarization term is represented as: $P^{(2)} = \epsilon_0\chi^{(2)}E^2$. Meep incorporates such nonlinear polarization terms into Maxwell's equations to simulate interactions between electromagnetic waves and the material in the time domain

$$\bigtriangledown \times H = \epsilon_0\frac{\partial E}{\partial t} + \frac{\partial P}{\partial t} \tag{2}$$

$$\bigtriangledown \times E = -\mu_0\frac{\partial H}{\partial t} \tag{3}$$

### B.2. Simulation Setup

The simulation domain consisted of a 100 $\mu$m medium, a 1 $\mu$m thick boundary layer, and 2 $\mu$m buffer regions at both ends. The spatial resolution was user-defined to capture fine electromagnetic field characteristics. Kerr media were placed at the center of the domain, with $\chi^2$ explicitly defined. The calculations were conducted using both FP32 and FP64 precision on single core of AMD Ryzen5 8500G.

### B.3. Source and Monitor Definition

The source was defined as a Gaussian plane wave with a central wavelength of 1.55 $\mu$m and a bandwidth of 0.15. Both the source and monitors were positioned 1 $\mu$m outside the nonlinear medium, with the electric field oscillating along the x-axis. Simulations were executed to allow sufficient decay of the fields after the source was turned off to confirm accurate measurements.

### B.4. Harmonic Generation and Analysis

Using the Meep's add flux function, the optical flux outside the nonlinear medium was measured, and the transmitted power spectra of the fundamental frequency ($\omega$) and harmonic components ($2\omega$, $3\omega$) were calculated. The add flux function records the time-domain values of electric and magnetic fields at specific locations, then performs a Fourier transform to convert them into the frequency domain to compute flux. This process allows precise analysis of the intensity of each frequency component within the user-defined frequency range and intervals. The analysis frequency range extended from $\omega$/2 to 3.5$\omega$, encompassing all relevant frequency bands of interest. Flux measurements were particularly useful for understanding the interaction between newly generated harmonic components and existing frequency components caused by the material's nonlinearity.

### B.5. Results and Reproducibility

Simulation results demonstrated how the intensity and distribution of harmonic components varied with changes in $\chi^2$. The nonlinear modeling capabilities of Meep enabled precise analysis of harmonic generation characteristics in nonlinear optical materials.

