# OpenReview forum: "Position: Significant impact of numerical precision in scientific machine learning"
_ICML.cc/2025/Position_Paper_Track — Submitted to ICML 2025 Position Paper Track_

### Official Review · Reviewer_zBvX · 2025-02-13

**Significance:** 3
**Argument Clarity:** 3
**Rating:** 3
**Confidence:** 4

**Questions:**

- What concrete benchmarks (reporting FP32 vs FP64) could be used?
- Could you elaborate more on collaboration with natural scientists?

**Discussion Potential:**

3

**Paper Summary:**

This paper advocates the need for considering high-precision (32 or 64 floating-point precision) in scientific ML, in order to address reliability and explainability of scientific discoveries.
The paper reports on experiments in classical scientific computing (density functional theory and finite difference time domain) and scientific ML to demonstrate the drawbacks of using single precision computation.

## update after rebuttal

The authors clearly emphasized in their rebuttal more insights about LLM and integration of ML into traditional methods. Based on this, I am now definitely convinced about keeping my initial score.

**Position:**

Yes

**Position In Title:**

Yes

**Related Work:**

3

**Strengths And Weaknesses:**

Strengths:

- the advocated position is supported by evidence in classical scientific computing and scientific ML
- it is likely to inspire discussion as scientific ML is a growing field
- the paper cites related works in an appropriate way

Weaknesses:

- the discussion around LLMs could be more elaborated
- same with the integration of ML into traditional methods

**Support:**

3

---

> ### Author Rebuttal · Authors · 2025-04-01
>
> We thank reviewer zBvX for the valuable and supportive feedback. In light of the comments provided, we have thoroughly reviewed and revised the manuscript to enhance its overall rigor and presentation.
>
> **A4-1: Discussion around LLMs and the integration of ML into traditional methods**
>
> Could the reviewer kindly provide more specific details about which aspects need further clarification? We will address these points comprehensively in the final response.
>
> **A4-2: Concrete benchmarks could be used**
>
> Similar to how computer vision and natural language processing fields evaluate methodologies using separate benchmarks for each specific task, we believe it is reasonable to repurpose existing benchmarks or datasets for each natural science research task. This is primarily because natural science fields and tasks are extensive to design separate benchmarks specifically for evaluating precision issues. Furthermore, when evaluating precision issues, it is essential to first verify that the model solves the given task above a certain threshold of performance.
>
> In this case, whereas previously we might have only compared the difference between ground truth and model prediction, we can now assess precision issues by additionally providing the difference between FP32 model predictions and FP64 model predictions. For example, most ML potential research published in the ML community uses the QM9 dataset to compare property prediction accuracy (*e.g.*, energy $E$). If we also provide information on the difference between properties predicted by FP32 models and those predicted by FP64 models (*e.g.*, $\Delta E_{FP64-FP32}$​), natural scientists can then determine whether an FP32 model is sufficient or if an FP64 model is necessary based on these values.
>
> **A4-3: Collaboration with natural scientists**
>
> Our intended meaning of "collaboration with natural scientists" is literally the need to collaborate with natural scientists from the relevant field when solving specific scientific tasks using ML. For example, if developing an ML model to predict celestial body orbits, collaboration with astronomy experts is necessary. While this may seem obvious, it is an aspect that people often overlook.
>
> We believe the most crucial domain knowledge ML researchers can gain through collaboration with natural scientists is understanding the accuracy requirements for a given task. For instance, in Reviewer tzuh's comment: *“you aren't using machine learning to understand those first 11 of 13 those decimal places (you already know them!); you are using to understand something about the last two, and how they are distorted in some crystal structure.”* However, without guidance from natural scientists, it is difficult to determine how many decimal places need to be estimated for a particular value, and how many are already known. Without this understanding of accuracy requirements, we continue to see papers published with claims such as "we achieved 98 % accuracy; therefore our model is excellent." while providing little actual benefit to natural science research. The discussion of numerical precision in our paper follows this same context. Only by comparing the predictions of 32-bit and 64-bit models, based on knowledge of the accuracy requirements for each scientific task, can we determine whether there is a scientifically significant issue.

---

> > ### Comment · Reviewer_zBvX · 2025-04-02
> >
> > I would like to thank the authors for their detailed reply.
> > At the end of Section 3, the authors mention "This inherent trade-off between computational efficiency and numerical precision highlights the importance of carefully designing LLM integration strategies in scientific applications to ensure both accuracy and practicality."
> >
> > 1) Could the authors provide more insights about the way of doing such careful design?
> >
> > Regarding the integration of ML into traditional methods, the authors only mention at the end of Section 4 that it is possible to use ML algorithms to provide initial guess/starting points for traditional algorithms. From the related works mentioned by the authors, such strategies have been already deployed.
> >
> > 2) Could the authors provide more insights about new strategies yet to be developed, and establish connection to the advocated position?

---

> > > ### Author Response · Authors · 2025-04-09
> > >
> > > We appreciate the reviewer’s thoughtful feedback, and here are our additional responses to the follow-up questions.
> > >
> > > **A4-4: More insights about “large language model (LLM)”**
> > >
> > > Performing training or inference with LLMs in double precision to address numerical precision concerns is highly inefficient in terms of both time and computational cost. A more practical approach would be leveraging existing LLMs while mitigating precision issues through targeted modifications.
> > >
> > > One promising method involves addressing tokenization problems with numerical values (where "3.11 is larger than 3.9" confuses LLMs). Instead of processing numbers as sequences of individual digit tokens, we could finetune LLMs to handle an entire numerical value as a single token. For output generation, dedicated prediction heads can be implemented to directly produce precise floating-point values, thereby bypassing the inaccuracies associated with converting numerical data to text tokens. Previous studies such as NumGPT [1] and xVal [2] demonstrate the efficacy of these approaches in numerically sensitive scientific applications.
> > >
> > > Beyond internal processing improvements, we can enhance the interface between LLMs and domain-specific models. Floating-point representations do not uniformly map real numbers. In other words, certain numerical ranges (e.g., between 0 and 1) are represented more densely than others (e.g., between 10 and 11). By applying normalization or regularization techniques that deliberately map model features into these densely-represented regions, we can maximize available precision. For example, the 3D MoLM [3] model (mentioned in our manuscript) utilizes a Q-Former module to connect a molecular encoder with an LLM by converting the encoder’s features into the LLM’s input token embeddings. Optimizing this interface with constraints that ensure converted token embeddings occupy the most precision-rich regions of floating-point representation could reduce numerical errors during cross-model information transfer.
> > >
> > > **A4-5: More insights about “integration of machine learning (ML) into traditional methods”**
> > >
> > > As noted by the reviewer, employing ML to generate initial guesses for traditional computational methods has been explored, though it is not yet widespread. Through our position paper, we hope to encourage more research in this direction. We plan to propose the following ideas for integrating ML with traditional computational methods in the revised manuscript:
> > >
> > > 1) The AI Scientist [4] demonstrated that LLMs can automate the entire ML research process from ideation to publication. Inspired by this, we envision LLMs independently solving scientific problems by autonomously identifying when simulations are needed, writing computation scripts, and analyzing results, all without human intervention. This approach is challenging since effectively using scientific simulation software requires graduate-level expertise, and comprehensive educational and instructional resources on such software remain scarce. Nonetheless, we believe this strategy is promising, as it leverages the strengths of LLMs’ scientific knowledge and reasoning capabilities combined with high accuracy provided by traditional computational methods, thus effectively avoiding precision issues.
> > >
> > > 2) While the first strategy focuses on combining LLMs with traditional methods, the second can be universally applied to any scientific ML model. Techniques such as Gaussian processes inherently provide predictions along with associated uncertainty estimates. If the uncertainty exceeds the accuracy threshold required for the specific scientific task, predictions can be rejected by using traditional methods instead. Extending this concept, we propose introducing ***interval arithmetic*** to evaluate and track uncertainties arising specifically from floating-point operations within ML models. Applying interval arithmetic to ML models allows predictions to be expressed as intervals rather than single values, reflecting the uncertainty accumulated during floating-point computations. When a predicted interval from the ML model does not satisfy the accuracy constraints necessary for the scientific task,  it clearly indicates that traditional computational methods are more appropriate for the given scenario.
> > >
> > > [1] Jin et al., “NumGPT: Improving Numeracy Ability of Generative Pre-trained Models.”, arXiv preprint (2021). \
> > > [2] Golkar et al., “xVal: A Continuous Number Encoding for Large Language Models.”, arXiv preprint (2023). \
> > > [3] Li et al., “Towards 3D Molecule-Text Interpretation in Language Models.”, ICLR (2024). \
> > > [4] Lu et al., “The AI Scientist: Towards Fully Automated Open-Ended Scientific Discovery.”,  arXiv preprint (2024).

---

### Official Review · Reviewer_h1Lj · 2025-03-08

**Significance:** 4
**Argument Clarity:** 3
**Rating:** 3
**Confidence:** 2

**Questions:**

1)  The paper argues for the use of FP64 in scientific ML, but what are the potential computational costs associated with this? Could the authors provide more details on how these costs might be mitigated (e.g., through mixed-precision training)?

**Discussion Potential:**

4

**Paper Summary:**

The paper argues that numerical precision, particularly the use of FP64 over FP32, is critical in scientific machine learning. It highlights that while the ML community has focused on computational efficiency through reduced-precision formats, higher precision is essential in scientific domains where small numerical differences can lead to significant inaccuracies. The paper presents evidence from density functional theory (DFT) and finite-difference time-domain (FDTD) simulations, showing that FP32 can lead to substantial errors in physicochemical properties and electromagnetic simulations. The authors also discuss the implications of numerical precision in ML potentials, physics-informed neural networks (PINNs), and large language models (LLMs). The paper concludes with actionable recommendations, including the exploration of high-precision models, benchmarking FP32 vs. FP64 results, and fostering collaboration between ML researchers and natural scientists.

**Position:**

Yes

**Position In Title:**

Yes

**Related Work:**

2

**Strengths And Weaknesses:**

$\textbf{Strengths:}$

1) Relevance and Importance: The topic is highly relevant to the ICML community, especially as ML is increasingly applied to scientific domains where precision is critical. The paper addresses a gap in the literature by focusing on the impact of numerical precision in scientific ML.

2) Evidence and Reasoning: The paper provides strong evidence through real-world examples and toy experiments, demonstrating the impact of numerical precision in DFT and FDTD simulations. The discussion on ML potentials and PINNs further supports the argument.

3) Clarity of Argument: The paper is well-structured and clearly argues its position. The authors effectively use examples and experimental results to illustrate their points.

4) Discussion Potential: The paper is likely to inspire constructive discussion within the ICML community, particularly among researchers working on scientific ML applications.

$\textbf{Weaknesses:}$

1) Lack of Counterarguments: The paper does not thoroughly address potential counterarguments, such as the computational cost of using FP64 or cases where FP32 might be sufficient. A more balanced discussion could strengthen the paper's position.

2) Empirical Evidence is (mostly non-ML): While the paper presents some empirical evidence from specific scientific domains with non-ML experiments, more extensive experiments or case studies across how ML applied is affected by precision errors in these different scientific domains could further validate the claims.

3) Lack of broader scientific experiments: The paper is only showcased two non-ML experiments on quantum chemistry and material science, although claims that precision plays big role in all ML for Science field. Maybe broadening the scope will strengthen the paper.

**Support:**

3

---

> ### Author Rebuttal · Authors · 2025-04-01
>
> We appreciate reviewer h1Lj’s insightful and constructive suggestions. We have responded to the comments in detail and made comprehensive revisions throughout the manuscript to reflect the feedback.
>
> **A3-1: Lack of counterarguments**
>
> We agree with the reviewer's comment. We will include the counterargument regarding computational cost in our revised manuscript, along with the alternative view suggested by Reviewer tzuh. In our subsequent response (A3-3), we will elaborate on our perspectives concerning computational cost and mixed-precision training. We will also continue to consider additional counterarguments during the remaining period, and if we develop any further insights, we will address these in our final response.
>
> **A3-2: Empirical evidence and lack of broader scientific experiments**
>
> First, we agree with the reviewer’s opinion and will supplement our content as thoroughly as possible. However, we would appreciate the reviewer’s understanding that the current manuscript primarily focuses on materials science, a field in which we have relatively more expertise, as we do not have comprehensive knowledge across all natural science disciplines. In particular, since many scientific simulation software packages require graduate-level training to use properly, we will focus on literature review to gather empirical evidence from other fields. We believe that as our position paper reaches researchers across various natural science disciplines, it will encourage systematic experiments that will offer more thorough evidence.
>
> **A3-3: Computational cost and mixed-precision training**
>
> First of all, for detailed explanations regarding computational cost, we would appreciate the reviewer’s reference to our response to Reviewer h6pn (A1-3). The key point is that the training and inference time with 64-bit models can be two to several dozen times higher than with 32-bit models. To mitigate these costs, we believe that mixed-precision training technique, which currently uses 16-bit and 32-bit precision together, should be further investigated for scenarios centered around 64-bit precision. For example, by analyzing the numerical sensitivity of each model component (layer), we could train sensitive layers using 64-bit precision while employing 32-bit or even 16-bit precision for less sensitive layers, thereby maximizing computational speed and reducing memory usage.

---

### Official Review · Reviewer_tzuh · 2025-03-15

**Significance:** 3
**Argument Clarity:** 4
**Rating:** 3
**Confidence:** 4

**Questions:**

In the Alternative View section, what about adding the position "I can always transform my problem such that, even though there are numbers in my problem that are high precision, the ML method is only working on small differences that are themselves low in precision. That is, any high-precision problem can be transformed by scalings (for units) and subtractions (of high precision base expectations) into a low-precision problem" ? Maybe the best argument against this alternative view is the idea of integration of, say, differential equations over many steps, and finite-difference schemes?

[Note added in rebuttal period: raising Significance score by 1]

**Discussion Potential:**

1

**Paper Summary:**

The paper argues that high-numerical-precision (and specifically 64-bit) is important for machine learning, and is especially important for ML in science. It argues that new ML methods should be developed and maintained at 64-bit, even if most users are at 32-bit.

## update after review
I raised my significance score; I would be happy to see this published in ICML.

**Position:**

Yes

**Position In Title:**

Yes

**Related Work:**

3

**Strengths And Weaknesses:**

Strengths:

I, myself (being a physicist) use 64-bit methods in ML all the time. The first lines in my code always switch jax to 64 bits. Thus I agree that 64-bit is critical for the sciences. (But see below.)

Weaknesses:

The position is not that strong or controversial. I think all ML-in-science types would probably agree? That said, I think that might be partly out of laziness. I can't imagine this paper creating a lot of controversy or discussion. Which is maybe a strength? Confused?

In principle, with enough effort and thought, any scientific problem can be done accurately at 32-bit. Of course I use my stuff at 64-bit, but it is because I am lazy. Even things like g-2 for the electron or muon, or the bond angle in some molecule, might be known to 8 or 13 decimals, but then all of the science can be done around *differences* away from the expectation, with proper computational techniques. After all, you aren't using machine learning to understand those first 11 of 13 those decimal places (you already know them!); you are using to understand something about the last two, and how they are distorted in some crystal structure.

All that said, I, like everyone else using ML, am lazy and I do indeed work at 64-bit so I agree, even if uncontroversially.

**Support:**

3

---

> ### Author Rebuttal · Authors · 2025-04-01
>
> We are grateful to reviewer tzuh for the thoughtful and encouraging comments. We have carefully addressed each point raised and have made substantial revisions to improve the clarity and quality of our manuscript.
>
> **A2-1: The position is not that strong or controversial**
>
> While developing our paper, we engaged in discussions about precision issues with various fields of colleagues. Interestingly, we received contrasting opinions depending on the research domain. When conversing with natural scientists, many shared the reviewer's position that using 64-bit precision is essential. On the contrary, when discussing with ML researchers, there was not only a lack of awareness regarding precision issues, but some even provided negative feedback on the claims presented in our paper.
>
> These negative opinions primarily centered on:
> 1) The assertion that scientific computing methods and ML methods operate differently internally, and therefore precision issues in the former cannot be definitively attributed to the latter.
> 2) The argument that even if precision issues exist, they may not be inherent to 32-bit models specifically, but rather indicate some deficiency in model architecture or methodology.
>
> We believe that this divergence of opinions further emphasizes the necessity of our paper. The lack of awareness about precision issues within the ML community suggests not that this topic is "non-controversial," but rather that it has not been sufficiently discussed. We hope that our paper serves not merely to raise an issue, but as a first step toward recognizing and seeking solutions for precision problems in the ML field.
>
> **A2-2: Alternative view**
>
> We appreciate the reviewer’s suggestion of an alternative view that we had not previously considered. The approach of predicting residuals has been widely utilized not only in natural sciences but also across various ML domains for solving diverse tasks, making it a valuable perspective. Similarly, the reviewer’s counterargument regarding differential equations over many steps and finite-difference approaches is compelling, as these methods are employed extensively beyond traditional scientific fields, including in applications such as diffusion models [ScoreSDE]. In the revised version, we will conduct further research on these topics and incorporate the alternative view the reviewer have proposed.
>
> [ScoreSDE] Song et al., Score-Based Generative Modeling through Stochastic Differential Equation. ICLR (2021).

---

> > ### Comment · Reviewer_tzuh · 2025-04-05
> >
> > I appreciate this response and I raised my significance score by 1.

---

> > > ### Author Response · Authors · 2025-04-09
> > >
> > > We appreciate the reviewer for carefully considering our response and for increasing the significance score by 1 point. We believe our work offers meaningful contributions to the field, and we are grateful for the reviewer’s recognition of its value.

---

### Official Review · Reviewer_h6pn · 2025-03-16

**Significance:** 4
**Argument Clarity:** 4
**Rating:** 3
**Confidence:** 4

**Questions:**

- More details around the setup for non-ML and ML experiments are needed (such as hyperparameters and such). The authors directly jump into the discussion of results.
- In the non-ML section, it might be good to also provide some examples around where precision doesn't matter as much, to help paint a more complete picture.
- Fig 4, "though broader testing needed", what does this mean? Either remove or elaborate this statement.
- "model trained entirely in FP32 from the outset might behave differently", quite handwavy, why not just check. This handwavy explanation is not in line with other more clear arguments that the authors make elsewhere, and is reducing the quality of their argumentation.
- Lines 323-325, "For example", it is quite difficult to parse the sentence, would be good to rephrase.
- "specialized FD method", acronym not introduced before usage.
- "Most ML research has primarily explored lower-precision formats", this does seem to be a true statement, but nevertheless a reference should be provided.
- One of the bigger downsides of the paper is that the authors do not explore what are the downsides of their proposal, and the negative impact. E.g., for higher precision there would be more latency and costs to training and running the models, and so on. There should be some discussion around that to make the paper more complete.

Typos:
- Line 161, the whitespace are strangely written. The authors should check the text throughout the paper to clean up such minor issues.
- "which an open-source"
- Line 220, "10'6", what is meant here?
- "retrosynethsis"

**Discussion Potential:**

3

**Paper Summary:**

The authors argue that the field of ML, and particularly the field of scientific ML, is not giving sufficient attention to the impact of numerical precision used in the experiments. They showcase a few non-ML approaches that are significantly impacted by change of precision, and then argue that ML has the same problem which they they investigate through literature overview and experiments on existing published models. Finally, they provide recommendations on how to move ahead and ensure the numerical precision is sufficiently taken into account.

## update after rebuttal
I appreciate the rebuttal comments, the authors do acknowledge the feedback and clarify how they are planning to address it. With that in mind, and after reading the other reviewers' comments which are quite aligned with mine, I would like to stay with the current recommendation.

**Position:**

Yes

**Position In Title:**

No

**Related Work:**

3

**Strengths And Weaknesses:**

The paper provides a clear message and good overview of the problem, together with very informative experiments both in the non-ML and ML domains. The provided recommendations to the field on how to move ahead are also well-supported and intuitive.
On the downside, the authors do not provide a sufficient overview of the areas where precision does not matter, which could be improved. Also, the ML impact argumentation is not as strong as in the case of non-ML.
Also, the authors do not explore the downsides of their proposals sufficiently.

**Support:**

3

---

> ### Author Rebuttal · Authors · 2025-04-01
>
> We sincerely appreciate reviewer h6pn’s positive and constructive feedback. In response to the reviewer’s valuable comments, we have provided detailed answers and thoroughly revised our manuscript accordingly.
>
> **A1-1: Examples where precision does not matter in non-ML section**
>
> During the preparation of this paper, we expected that the main readership would consist of ML researchers. Therefore, we believed that to present our argument convincingly, it was essential to clearly demonstrate the existence of precision issues in traditional scientific computing. As a result, we focused on illustrating examples where precision problems occur, overlooking the possibility that this approach might appear biased. We accept the reviewer's feedback and will incorporate examples where precision issues do not matter in the revised manuscript.
>
> Specifically, we will include additional case studies of FDTD simulations examining SiO$_2$ absorption under applied electric field conditions. Our analysis revealed no significant discrepancies between FP32 and FP64 implementations, indicating that certain empirical computational results are not significantly affected by numerical precision.
>
> **A1-2: ML impact argumentation is not as strong as in the case of non-ML**
>
> Given that natural science encompasses a broad range of disciplines, we were concerned about making overclaims during the writing process. Therefore, to avoid the problem of overclaiming, we decided to present our ML-related content with relatively modest assertions. If the reviewer thinks we could make stronger claims, we are willing to accept that feedback and revise our content accordingly.
>
> Meanwhile, the reviewer's critique regarding the direct comparison between models trained with 32-bit and 64-bit precision is valid, and we will supplement additional related results in the field of ML impact to support the importance of precision considerations in practical scientific ML research. While non-ML experiments can be done under predetermined scenarios, ML experiments require further consideration of various factors including selection of training and evaluation datasets, design of appropriate model architectures, hyperparameter tuning, and computational resource constraints. Given these practical limitations, we chose to indirectly demonstrate precision issues through examples of pretrained models. We thought this approach could effectively illustrate precision issues, but it seems this was not sufficient.
>
> **A1-3: Negative impact**
>
> We overlooked a simple yet crucial counterargument. The use of higher precision operations clearly demands more computational resources. For instance, the official datasheet of the NVIDIA H100 SXM GPU indicates that FP64 operations achieve 34 TF, while FP32 operations reach 68 TF, twice the performance. This relationship suggests that training models at 64-bit precision would require approximately double the computational time compared to 32-bit implementations.
>
> However, it would be incorrect to assume that 64-bit floating-point operations always demand only twice the resources of 32-bit operations across all hardware. The un-official datasheet of the NVIDIA RTX A6000 (Ampere) GPU indicates that FP64 performance is merely 1/64 of the FP32 capability (the official datasheet does not explicitly state FP64 performance). This unexpected performance difference suggests that using 64-bit precision could potentially increase training and inference time by a factor of 64. This substantial computational performance gap, even setting aside the fact that memory usage doubles, presents a major practical challenge for the widespread use of 64-bit precision in scientific ML. Taking these into account, we will revise the manuscript to include a comprehensive discussion on the practical limitations and negative impacts of using 64-bit precision in scientific ML.
>
> **A1-4: Details around the setup for non-ML and ML experiments**
>
> Acknowledging the reviewer's feedback, we will provide a more comprehensive description of our experimental setup. We will supplement the content currently described in the appendix and also address it in the main manuscript.
>
> **A1-5: “Most ML research has primarily explored lower-precision formats”**
>
> In the revised version, we will include proper reference to all relevant studies.
>
> **A1-5: "though broader testing needed"**
>
> As recommended by the reviewer, we will remove this phrase.
>
> **A-6: Readability and typos**
>
> We sincerely appreciate the reviewer's thorough reading of our paper, and we will address all the points the reviewer has raised. Thanks to the reviewer’s feedback, the quality of our manuscript has significantly improved.

---

> > ### Comment · Reviewer_h6pn · 2025-04-05
> >
> > I appreciate the rebuttal comments, the authors do acknowledge the feedback and clarify how they are planning to address it. With that in mind, and after reading the other reviewers' comments which are quite aligned with mine, I would like to stay with the current recommendation.

---

> > > ### Author Response · Authors · 2025-04-09
> > >
> > > We sincerely thank the reviewer for the positive feedback on our rebuttal and for thoughtfully maintaining the recommendation after carefully considering both our responses and the comments from other reviewers.

---

### Decision · Program_Chairs · 2025-04-30

**Decision:**

Reject (with encouragement)

**Comment:**

Your paper was favorably reviewed, and we would have liked to accept it this year.  However, due to constraints on conference capacity, we had to reject some papers despite positive reviews.  The area chair's original meta-review (copied below) is a testament to the strengths of your paper.

----
I agree with the authors that the predominant view in the ML community is that FP32 is sufficient for most tasks. Therefore, the current paper clearly fits within the scope of the position track.

I would make the distinction between
 * Claim A: the inputs / outputs to scientific ML models require high precision; and
 * Claim B: the internal computation of (at least some) scientific ML models require high precision.

Generally, at the highest level, we are interested in whether a crystal structure is stable or not, whether it has the desired property or not, etc and I don't believe model predictions at this level require a high precision.

I can see that Claim A may be true e.g., if the ML model is used as an energy/force provider to a traditional numerical integrator. However, it can easily be side-steped if we could train a model that predicts the output of the numerical integrator directly.

Therefore, in my view the more interesting discussion is Claim B, whether a ML model should use high precision computation *internally* to be a useful scientific model. In this context, non-ML models provide a useful data point that if we are allowed to use high precision we can make useful predictions, but it doesn't prove that it is impossible to create a similarly accurate model that only uses low precision computation.

PINNs are interesting because e.g., finite difference calculation involved in PINNs are likely to be more susceptible to errors from low-precision training. However, this would also not prove that there is no other network that is as accurate as PINNs but is less susceptible to numerical errors.

I think the paper would be much stronger if the authors could present a ML task (even a toy one) that is hard for FP32 models but somewhat easier for a FP64 model. My guess is that this is unlikely to be the case as long as we use common building blocks like linear or attention layers, but I could be wrong.